# Synthesis, Structural Characterisation, and Electrochemical Properties of Copper(II) Complexes with Functionalized Thiosemicarbazones Derived from 5-Acetylbarbituric Acid

**DOI:** 10.3390/molecules29102245

**Published:** 2024-05-10

**Authors:** Alfonso Castiñeiras, Nuria Fernández-Hermida, Isabel García-Santos, Lourdes Gómez-Rodríguez, Antonio Frontera, Juan Niclós-Gutiérrez

**Affiliations:** 1Department of Inorganic Chemistry, Faculty of Pharmacy, University of Santiago de Compostela, 15782 Santiago de Compostela, Spain; alfonso.castineiras@usc.es (A.C.); lurdesgomezrodriguez@edu.xunta.es (L.G.-R.); 2Department of Chemistry, University of Illes Balears, Crta de Valldemossa km 7.5, 07122 Palma de Mallorca, Spain; toni.frontera@uib.es; 3Department of Inorganic Chemistry, Faculty of Pharmacy, University of Granada, 18071 Granada, Spain; jniclos@ugr.es

**Keywords:** thiosemicarbazone complexes, 5-acetylbarbituric derivatives, crystal structures, cooper(II) complexes, electrochemical analysis

## Abstract

The reaction between 5-acetylbarbituric acid and 4-dimethylthiosemicarbazide or 4-hexamethyleneiminyl thiosemicarbazide produces 5-acetylbarbituric-4-dimethylthiosemicarbazone (H_2_AcbDM) and 5-acetylbarbituric-4N-hexamethyleneiminyl thiosemicarbazone (H_2_Acbhexim). Eight new complexes with different copper(II) salts have been prepared and characterized using elemental analysis, molar conductance, UV–Vis, ESI-HRMS, FT-IR, magnetic moment, EPR, and cyclic voltammetry. In addition, three-dimensional molecular structures of [Cu(HAcbDM)(H_2_O)_2_](NO_3_)·H_2_O (**3a**), [Cu(HAcbDM)(H_2_O)_2_]ClO_4_ (**4**), and [Cu(HAcbHexim)Cl] (**6**) were determined by single crystal X-ray crystallography, and an analysis of their supramolecular structure was carried out. The H-bonded assemblies were further studied energetically using DFT calculations and MEP surface and QTAIM analyses. In these complexes, the thiosemicarbazone coordinates to the metal ion in an ONS-tridentate manner, in the O-enolate/S-thione form. The electrochemical behavior of the thiosemicarbazones and their copper(II) complexes has been investigated at room temperature using the cyclic voltammetry technique in DMFA. The Cu(II)/Cu(I) redox system was found to be consistent with the quasi-reversible diffusion-controlled process.

## 1. Introduction

It has been known for more than 30 years that climate change threatens the Earth’s well-being. Therefore, since the Berlin Climate Change Conference in 1995, regular assessments based on the 1997 Kyoto Protocol and the 2015 Paris Agreement have been carried out and a course of action has been proposed. The importance of energy and environmental issues in science and society implies an effort in the development of green energy devices using organic optoelectronics. Compared to conventional inorganic semiconductors, organic and metal–organic small molecules have a higher degree of versatility and can be synthesized using different strategies to offer low-cost mass production in a wide range of applications, such as organic light-emitting diodes, energy storage, hybrid photovoltaic cells, and artificial photosynthesis. These properties have driven the rapid development of the synthesis of new transition metal complexes, generating enormous interest for different energy-related applications [1].

Coordination compounds play a critical role in modern inorganic and bioinorganic chemistry. They are considered the backbone of various chemical industries [2] due to their wide application in different areas such as metal detection, bioimaging, drug delivery, chemosensors, and pharmacological and medicinal chemistry. Furthermore, the application of green chemistry and the development of biotechnological and organocatalytic methods leads to environmentally friendly chemical technologies [3].

Because of their utilitarian properties and interesting structures, coordination com-pounds have attracted increasing interest. The rapid development of materials science and crystal engineering has greatly encouraged the use of complexes as functional materials, such as photocatalysts, supercapacitors, nonlinear optical materials, porous materials, and biomedical applications [4]. From this perspective and from the point of view of sustainability, the metal complexes used in these applications must be manufactured from sufficiently abundant materials and, in order to enter mass market applications, cost-effective metals with a d^10^ orbital configuration—such as Cu(I)—have to be, and currently are, the center of numerous research studies.

Copper, one of the cheapest and most abundant elements on Earth, is a good choice for developing inexpensive and efficient coordination complexes for all energy-related applications and to replace other unsustainable and toxic metals (Co, Pt, Ir, and Ru). There is a great need for further development of copper coordination complexes to increase the performance of each energy-relevant application [1]. In addition, copper complexes constitute an important class of molecules from several points of view: bioinorganic, catalysis and magnetism. Copper is an essential bioelement, responsible for numerous catalytic processes in living systems, where it presents different nuclearities and is present in the enzymes of biological systems to perform their function through redox reactivity [5].

The design of new ligand systems depends on their metal reactivity and the binding modes. In the absence/existence of coordinated anions or small molecules, the stability in solution and characteristics of the formed fragments can give ligand aggregates [6] and they are transcendental for their existence in solution.

Redox-active ligands are easily oxidized or reduced, compared to classic spectator ligands, extending the redox reactivity of metal atoms. Normally, the number of electrons that a mononuclear metal complex can transfer to a substrate is limited to one or two, but in complexes with active redox ligands, the ligands can provide additional electrons [7].

Schiff bases have been studied extensively because of their fundamental role in the coordination chemistry of transition metals and major groups, their simple preparation method and structural variety, and their innumerable applications as chelating ligands, as catalysts, as dyes, as initiators in polymerization, and as luminescent compounds. Biologically, they have also been tested as antibacterial, antifungal, antitumor, and antiviral agents including insecticides [8].

One group of Schiff bases that have attracted a great deal of research attention due to their medical application in the treatment of such feared diseases as leprosy and tuberculosis in the 1950s are thiosemicarbazones (TSCs) and they also have antiviral, antibacterial, anticancer, anti-inflammatory, antimalarial, and anti-HIV activity [9,10]. They can also be used for metal analysis, device applications related to telecommunications, optical computing, storage, and information processing [11]. TSCs are synthesized by the condensation reaction of an aldehyde or ketone with a thiosemicarbazide, being found as thiol-thione tautomers; although in some cases, especially when they are coordinated, the thiol form can be deprotonated to form thiolate anions [12]. In general, TSCs coordinate with metals in a bidentate manner through thiocarbonyl sulfur and azomethine nitrogen, mainly in a square-planar geometry. However, they can also be coordinated in a tridentate or tetradentate manner, among other possibilities, when the aldehyde is functionalized or it is a bis-thiosemicarbazone. Clearly, the biological activity of the resulting complexes will be affected by the type of metal center and functionalization [13].

In previous studies, we have reported on the synthesis and characterization of some novel thiosemicarbazones derived from 5-acetylbarbiturate acid [14] and its Ni(II), Pd(II), and Pt(II) complexes [15,16]. In the work described here, we use the potentially tridentate ligands 5-acetylbarbiturate-4-dimethylthiosemicarbazone (H_2_AcbDM) and 5-acetylbarbiturate-4N-hexamethylenenimininilyl thiosemicarbazone (H_2_Acbhexim) (Figure 1), in whose abbreviation “H_2_” denotes its two potentially dissociable acid protons, i.e., the barbiturate proton (enolic) and the hydrazine proton (thiol). Generally, these thiosemicarbazones are expected to bind to a metal center as dianionic tridentate O,N,S donors (a binding mode we have observed in platinum complexes), as monoanionic tridentate O,N,S donors, after the loss of the enolic barbiturate proton, as thiolate ligands, after dissociation of the hydrazine proton N–H, or as neutral ligands. In this study, we report a series of Cu(II) mononuclear complexes with H_2_AcbDM and H_2_Acbhexim. The eight newly prepared complexes were characterized experimentally, using UV–Vis FT-IR, EPR spectral analysis, cyclic voltammetry, magnetic susceptibility, conductivity measurements, and DFT theoretical studies.

## 2. Results and Discussion

### 2.1. Synthesis and Characterization

The complexes synthesized at ambient temperatures under the magnetic stirring of ethanolic solutions containing thiosemicarbazone ligands and copper salts in a 1:1 molar ratio gave air-stable solids in a good yield. The precipitates formed were then collected by filtration and washed thoroughly with ethanol. The elementary analysis is concordant with the formulas [Cu(HAcbDM)(OAc)]·3H_2_O (**1**·3H_2_O), [Cu(HAcbDM)Cl]·1/2EtOH (**2**·1/2EtOH), [Cu(HAcbDM)(NO_3_)] (**3**), [Cu(HAcbDM)(H_2_O)_2_]ClO_4_ (**4**), [Cu(HAcbHexim)(OAc)]·5H_2_O (**5**·5H_2_O), [Cu(HAcbHexim)Cl]·1/2EtOH (**6**·1/2EtOH), [Cu(HAcbHexim)(NO_3_)]·1/2H_2_O (**7**·1/2H_2_O), and [Cu(HAcbHexim)ClO_4_] (**8**); crystals suitable for structural characterization by X-ray diffraction were only obtained from those with the following formulas: [Cu(HAcbDM)(H_2_O)_2_](NO_3_).H_2_O (**3a**), [Cu(HAcbDM)(H_2_O)_2_]ClO_4_ (**4**), and [Cu(HAcbHexim)Cl] (**6**).

The complexes showed a good solubility in acetonitrile, dimethylformamide (DMF), and dimethyl sulfoxide (DMSO), and partial solubility in acetone and chloroform. However, they are insoluble in ether, methanol, and ethanol. The mass spectra of the complexes (FAB) (Appendix A) in some cases show signals for the molecular ion peak, although the most significant peaks correspond to the fragments [Cu(L)]^+^ and [L]^+^ at *m*/*z* 334 and 272 for **1**–**4**, and at *m*/*z* 388 and 326 for compounds **5**–**8**, respectively. These fragmentation patterns involve the addition or loss of a hydrogen atom from the fragments.

The molar conductivities of freshly prepared solutions of the complexes in DMF are in the range 10–35 Λ_m_/S cm^2^ mol^−1^, which indicates that the compounds are neutral. However, [Cu(HAcbDM)(H_2_O)_2_]ClO_4_ is a 1:1 electrolyte (83 Λ_m_/S cm^2^ mol^−1^). The molar conductivity value for [Cu(HAcbDM)(NO_3_)] is of 54 Λ_m_/S cm^2^ mol^−1^, and this value may correspond to a mixture of [Cu(HAcbDM)(NO_3_)] (neutral) and [Cu(HAcbDM)(H_2_O)_2_](NO_3_)·H_2_O (1:1 electrolyte) present in the solution [17].

### 2.2. FT-IR Spectra

Spectra in the mid-infrared region (Appendix A) have bands in the range of 3500–3100 cm^−1^ associated with ν(NH) and ν(OH) of the coordination water molecules and crystallization solvent molecules. The coordination of copper(II) ligands causes shifts in the bands at low frequencies, due to ν(CN) + ν(CC), in the range of 1625–1520 cm^−1^ in the free thiosemicarbazones. The band due to ν(CO), which in ligands appears as a strong band, around 1725 cm^−1^ in complexes, is shifted to 1700–1710 cm^−1^. In addition, the band corresponding to the C=S stretching modes, which in ligands is observed between 830 and 840 cm^−1^, appears at lower frequencies in complexes. The existence of the perchlorate group is confirmed by two intense bands [18] around 1120 and 625 cm^−1^, while the appearance of two bands in the range of 1350–1280 cm^−1^ is due to the presence of nitrate groups [19]. In the spectra of acetate complexes, the characteristic bands ν_a_ (COO^−^) and ν_s_ (COO^−^) are observed in the range between 1540 and 1515 cm^−1^ and between 1420 and 1400 cm^−1^, respectively. The value of Δ [the separation between the two components of ν(COO^−^), Δ = ν_a_(COO^−^) − ν_s_(COO^−^)], is in the range of 90–130 cm^−1^, these data being consistent with the mode of coordination of the acetate anion in the complexes [20,21]. This behavior is due to the monodeprotonation of thiosemicarbazones by N_azo_-H, and ligand coordination via ONS. This coordination by the carbonyl oxygen atom, the azometin nitrogen atom, and the thiolate sulfur atom is also supported by the position of the bands ν(Cu-O) at 550 cm^−1^, ν(Cu-N) at 425–450 cm^−1^, ν(Cu-S) around 315 cm^−1^, and ν(Cu-Cl) at 250 cm^−1^ [22] (Appendix A).

### 2.3. UV–Visible Spectra

The electronic spectra of copper(II) complexes have been studied qualitatively in the solid phase (Appendix A). The spectra of all complexes show different bands in the UV region. A high-energy band appearing between 28,400 and 30,400 cm^−1^ was assigned to intraligand charge transfer transitions (ILCT), π→π*, while another observed in some complexes between 21,700 and 22,100 cm^−1^, was attributed to ligand-to-metal charge transfer transitions (LMCT), O→Cu^II^. In the visible region, the spectra show two distinct bands that have been assigned to ^2^E_g_→^2^T_2g_ transitions. In Complexes **2**, **6**, and **8**, both bands appear between 15,450 and 16,500 cm^−1^ and between 14,350 and 14,425 cm^−1^ and fall in the range of 16,650–14,250 cm^−1^, usually typical of square plane complexes. In Complexes **1**, **3**, **5**, and **7**, one band appears between 17,390 and 17,640 cm^−1^ and the other between 14,150 and 15,650 cm^−1^, suggesting that these complexes have a distorted square pyramidal coordination geometry [23].

### 2.4. Magnetic Susceptibilities

The magnetic moments of mononuclear copper(II) complexes are generally observed in the range 1.7 to 2.2 µ_β_, regardless of their stereochemistry. In most of the complexes studied here, the magnetic moments are in the range of 1.85–2.25 BM, which confirms the presence of mononuclear Cu(II) [24], because those of Cu(I) are expected to be diamagnetic. However, [Cu(HAcbHexim)ClO_4_] presents a μ_eff_ value of 1.51 MB, which may be due to the existence of interactions between the Cu(II) centers of two neighboring molecules.

### 2.5. EPR Analysis

The EPR spectra of Compounds **1**–**8** (Appendix A) in the polycrystalline state performed at room temperature show different types of geometric species and are similar to others previously reported for Cu^II^ complexes with thiosemicarbazones [25]. Compounds **1** and **5**–**7** present isotropic spectra containings a single broad signal with values for g in the range of 2.108–2.081. On the other hand, the spectra of Compounds **2**–**4** and **8** present the characteristics of axial anisotropy with well-defined values of g_II_ and g_⊥_ around 2.189–2.372 and 2.043–2.095, respectively, where the value of g_II_ > g_⊥_ > 2.0023, which supports that d_x2−y2_ is the fundamental term in a square-planar geometry. According to Hathway and Billing [26], the geometrical parameter G is used to measure the exchange interaction between copper centers, which for axial spectra can be calculated using the following equation: G_(axial)_ = g_II_ − 2.0023/g_⊥_ − 2.0023; so if the value is between 3 and 5, it indicates that the fundamental state of these complexes is d_x2−y2_ [27], and this agrees with the intermediate structure between tetrahedral and square planar. Coordination number of four in the complexes is also verified by the relatively low g_II_ values, indicating strong coordination and a significant degree of covalence in the bonds. In addition, a slight increase in g_II_ has been observed when the volume of substituents on thioamide nitrogen increases, as occurs in similar complexes.

### 2.6. Molecular Structures and Supramolecular Analysis

Table 1 summarizes the relevant crystal data and the refinement of the structures of Compounds **3a**, **4**, and **6**. Table 2 shows the coordination bond lengths and angles, and Appendix A shows the parameters of the hydrogen-bonding interactions of the three compounds. The asymmetric unit of each of the compounds is represented in Figure 1, respectively. In the asymmetric unit of **3a** (Figure 1a), there are two independent complex molecules (I and II) and the coordination number is five due to the [HAcbDM]^−^ ligands, which behave as mono-deprotonated tridentate ONS, and the enolate oxygen atoms, azomethine nitrogen and thiolate sulfur. The asymmetric unit of **4** (Figure 1b) is formed by a single complex molecule that is geometrically similar to either of the two **3a** molecules, but, in this case, with [HAcbHexim]^−^ as the ligand.

In both compounds, the two remaining coordination positions are occupied by the oxygen atoms of two water molecules, originating a cationic complex, where the electroneutrality of the molecule is compensated by an anion nitrate (**3a**) or perchlorate (**4**). In addition, the **3a** asymmetric unit contains a crystallization water molecule for each complex molecule. In the two complexes, the coordination geometry around Cu(II) can be described as from square pyramidal to square pyramidal slightly distorted, according to the Addison parameter τ_5_ of 0.07 (molecule I of **3a**), 0.24 (molecule II of **3a**), and 0.02 (**4**) [τ = (β − α)/60) where α and β are the two largest angles L-M-L′; τ is 1 for a perfect trigonal bipyramid and is 0 for a perfect square pyramid [28]. For **3a** and **4**, the three atoms of the ligand and the oxygen atom of a coordination water molecule occupy the four corners of the basal plane of the pyramid, at an average distance of 1.936 for Cu-O or Cu-N and 2.247 Å for Cu-S (Table 2), while the oxygen atom of the other coordination water molecule occupies the axial positions at greater bond distances from 2.441 (4) and 2.477 (4) Å (**3a**) and 2.609 (10) Å (**4**) (Table 2).

Single crystals of [Cu(HAcbDM)(H_2_O)_2_](NO_3_)·H_2_O (**3a**), [Cu(HAcbDM)(H_2_O)_2_]ClO_4_ (**4**), and [Cu(HAcbHexim)Cl] (**6**) suitable for X-ray diffractometry were obtained from the stock solution by slow evaporation at room temperature.

In the crystal structure of **6**, the asymmetric unit contains one molecule of the complex (Figure 1c). As in the structures described above, the mono-deprotonated thiosemicarbazone ligand is coordinated to the metal in the expected terdentate manner via the thiolate sulfur S(1), the azomethine nitrogen N(12), and the enolate oxygen O(11), creating five- and six-membered chelate rings with N–Cu-S and O–Cu-N bite angles of 87.50 (6) and 87.90 (8)°, respectively. The dihedral angle between the mean planes of the two chelate rings are 14.228 (2)°, which contrasts with the values of 8.46 (1), 7.77 (1), and 6.8 (2)° found in the **3a** and **4** molecules. The Cl^−^ ion is coordinated to the metal in the trans position with respect to the donor nitrogen atom. The Cu(II) is thus nested in an ONSCl core with circum-metallic bond parameters that deviate slightly from ideal square-planar geometry.

This description is in agreement with the value of 0.06 calculated for *τ*_4_, the index for the four-coordinate complexes, which is the sum of angles α and β—the two largest theta angles in the four-coordinate species—subtracted from 360° all divided by 141°, where *τ*_4_ is 1.00 in tetrahedral geometries and 0.00 in square-planar geometries [29]. This core is nevertheless essentially planar, with a mean plane where the maximum deviation is 0.043 (1) Å in N(12), and from which the metal atom deviates 0.0641 (1) Å and forms a dihedral angle of 13.83 (9)° with the thiosemicarbazone moiety. The Cu-O, Cu-N, and Cu-S distances are all within the range that has been observed previously in **3a** and **4** complexes (Table 2). The fact that the Cu-S(1) distance [2.2311 (7) Å] is shorter than the distances of Cu-S in **3a** and **4** [2.2487 (14) and 2.247 (3) Å, respectively], and somewhat shorter than the Cu-Cl(1) distance [2.2338 (7) Å], shows that the trans-influence of the imine nitrogen atom is greater than that of the keto oxygen atom. In the thiosemicarbazone, the C–S(1) and C–O(11) bond lengths [1.714 (2) and 1.265 (3) Å, respectively] attest to the thiolate character of S(1) and the keto character of O(11), but the differences in bond lengths and bond angles with respect to the other complexes [greater for C–S and minor for C–O(11)], explains lower planarity in the thiosemicarbazone. The dihedral angle between the thiocarbazide moiety and the barbiturate ring is just 22.3 (1)°, versus 4.9 (3), 13.1 (3), and 7.6 (3)° in **3a** and **4**.

Unlike the results detected in the 5-acetyl barbituric-based free thiosemicarbazones, where many of their conformational parameters depend on the substituents present on nitrogen thioamide, in the complexes, these parameters are very similar [14]. When free, 5-acetylbarbituric hydrazine-1-carbothioamide (H_2_AcbNH_2_) is planar and, when N-mono and disubstituted, the plane of the 2,4,6-pyrimidinetrione ring is rotated between 46 and 57° with respect to that of the thiosemicarbazone moiety. However, in the complexes studied here, the plane of the 2,4,6-pyrimidinetrione ring forms angles of 4.8 (3)/13 (3)° at **3a**, 8.2 (6)° at **4**, and 23.4 (1)° at **6**. Obviously, this deviation from the values found in the free ligands is imposed by the square-planar coordination geometry of the complexes.

On the other hand, it is known that the hydrazine nitrogen atom in the thiosemicarbazone N12-H13-C18 fragment loses the sp^2^ hybridization planarity due to a hysterical impediment with the hydrogen atoms of the substituents on the thioamide nitrogen [14]. In complexes, this loss is maintained so that the N13 atom is outside the plane formed by N12-H13-C18 between 0.13 (3) and 0.33 (2) Å for molecules I and II of **3a**, respectively, 0.28 (8) Å in **4**, and 0.17 (1) Å in **6**.

Finally, the bond lengths and bond angles of the 2,4,6-pyrimidintrione ring in all three complexes are in the ranges allowed for barbituric acid derivatives [30].

In relation to supramolecular packing, and more specifically to hydrogen-bond formation, in the complexes all thiosemicarbazones have two H-acceptor oxygen atoms in common, located in the 2,4,6-pyrimidinetrione ring, and three N–H donors, two in the aforementioned ring and a third that corresponds to the hydrogen atom on the hydrazine nitrogen of the thiosemicarbazone moiety.

In the crystal packing of **3a**, each cation forms a hydrogen bond with one of the oxygen atoms of a nitrate with N13-H13 as a donor, and another with a crystallization water molecule through the O14-H14A bond of the coordination water molecule that occupies the axial position. In turn, the crystallization molecule and nitrate form a new hydrogen bond O1-H1B···O43, forming a heterosynthon with an $R_{3}^{3}$(11) graph set, where the nitrate ion functions as a bidentate bridge (Figure 1a).

In addition, there are other hydrogen bonds that govern packing. Each HAcb4DM ligand is linked to two other nearest neighbors by means of N–H···O bonds, forming two supramolecular heterosynthons with an $R_{2}^{2}$(8) graph set in which the two HN-C(O) groups of the 2,4,6-pyrimidinetrione ring participate, generating planar zig-zag chains in the direction of the bisector of the angle between the b and c axes. Also, in the same direction, nitrate ions participate in new hydrogen bonds as triple bridges forming new heterosynthons with graph sets $R_{3}^{2}$(8), $R_{4}^{3}$(14), $R_{6}^{6}$(20), and $R_{4}^{4}$(21) (Figure 2). This creates a robust 3D network in which all the possible hydrogen-bond donors and acceptors contained in the molecules participate (Appendix A).

The crystal packing of **4**, based on the formation of hydrogen bonds (Appendix A), includes dimer units of cations in inverted positions, one relative to the other, linked by hydrogen bonds between the coordination water molecules (Figure 3a) forming a heterosynthon with an $R_{2}^{2}$(8) graph set where the basal water molecule acts as the donor and the axial as the acceptor. These dimers in turn bind to new inverted dimers via thiosemicarbazone ligands with the formation of the $R_{2}^{2}$(8) graph set heterosynthon between NH-C(O) groups described in Compound **3a** packaging (Figure 3b).

This association is reinforced by a hydrogen bond between an O2-H of the water molecule in the axial position and the oxygen atom O13 in Position 4 of the 2,4,6-pyrimidinetrione ring, forming a new supramolecular heterosynthon $R_{3}^{2}$(10) and giving rise to ladder-like cation chains parallel to the *c*-axis (Figure 3b). These chains are held together in planes parallel to the ab plane through perchlorate bridges connected by means of new hydrogen bonds, where the second O–H of each water molecule in the axial position of one chain and the water molecule in the basal position of the neighboring chain participate as donors, and the O16 and O18 atoms of the perchlorate anions as acceptors (Figure 4a). Corrugated sheets similar to the arrangement on a tile roof are then formed, propagating parallel to the *b*-axis (Figure 4b).

The crystal packing of this compound is also aided by chelated ring-chelate···ring-chelate stacking interactions [31] between the five- and six-membered chelate rings that form the thiosemicarbazone ligand in coordination (Appendix A), with inter-centroid distances of 3.715–3.853 Å (Appendix A). Also related to these interactions, self-assembled dimers present metal···chelate ring type interactions with a Cu-centroid distance of 3.911 Å (Appendix A).

In Compound **6**, only three intermolecular hydrogen bonds of the N–H···O type define crystal packing (Appendix A). An N–H corresponding to the hydrazine nitrogen atom of the thiosemicarbazone moiety (N13) acts as a donor against the carbonyl oxygen atom (C13) at Position 4 of the 2,4,6,-pyrimidinetrione ring of a neighboring molecule (Figure 5a), forming a zig-zag chain running along the *a*-axis (Figure 5b). In addition, the two hydrogen atoms on the pyrimidine nitrogen atoms are involved in two other hydrogen bonds with the other carbonyl oxygen atom (O15) and with the chloride ligand (Cl1) of the two new nearest neighboring molecules (Figure 5c), giving rise to supramolecular heterosynthons with an $R_{2}^{2}$(10) graph set in planar chains running parallel to the *b*-axis (Figure 5d). In this way, the conjunction of both chains gives rise to a dense 3D network (Appendix A).

### 2.7. Cyclic Voltammetry Studies

Systematically, the cyclic voltammograms of Complexes **1**–**4** and **6**–**8** were carried out over a wide range of potential to show all the peaks. These measurements were performed by delimiting the areas where individual peaks appeared to investigate possible connections between the associated electrochemical reactions (Appendix A).

In the cyclic voltammograms (Figure 6a), first, two systems of peaks, I and II, were observed with the cathodic form and absence of the anodic form in the majority of the cases. At more negative potentials, one or two cathodic peaks (III and IV) without the corresponding anodic peaks were registered. Finally, at potentials more positive than those of system I, an anodic peak (V) was observed without the corresponding cathodic form. In Complex **3**, additional minor peaks were observed as shown in Figure 6b.

All complexes with peak I_a_ present the typical form of an electrode process of a reagent adsorbed on the electrode. In general, using a more negative vertex potential of 1.1 V brings about the disappearance of I_a_, even with long times of delay (Figure 6a), indicating that it is potentially at this point that adsorption occurs significantly.

System II is very close to the previous one, so that the parameters of II_c_ can only be estimated approximately. This proximity is an indication of both peak systems are generated by the same type of electrochemical reaction taking place with molecules of slightly different structures. The fact that II_a_ does not appear is indicative that the corresponding oxidation reaction is extremely slow or even non-existent.

The position of both systems indicates the reduction and reoxidation of Cu^II^ according to the following scheme [32]:I_c_: LCu^2+^ + e^−^ → LCu^+^, potential range: −0.220/−0.320 V
I_a_: LCu^+^ → LCu^2+^ + e^−^, potential range: −0.017/0.087 V
II_c_: L′Cu^2+^ + e^−^ → L′Cu^+^, potential range: −0.562/−0.764 V
II_a_: L′Cu^+^ → L′Cu^2+^ + e^−^, potential range: −0.327/−0.422 V

The characteristics of I_a_ prove that the difference between the structures of the complexes giving rise to I or II affects the adsorption of their reduced forms on the electrode surface. The almost planar geometry of the ligand would justify the existence of a certain adsorption capacity of ligand and the fact that it only affects the reduced form of one of the variants of the complex (resulting in peak Ia) indicates the existence of an additional factor that would determine the above-mentioned adsorption [33].

In Complexes **1** and **7**, the fact that the cyclic voltammograms are carried out from very positive initial potentials (2 V vs. ECS) favours the appearance of I_c_ to the detriment of II_c_. This behaviour indicates that L and L′ can represent oxidized and reduced forms of some component of the ligand, V_a_ probably being the peak resulting from the electrochemistry oxidation of L′Cu^2+^ to LCu^2+^. On the other hand, the appearance of this peak when a voltammogram is performed in a potential region restricted to the range in which it appears (Figure 6c,d) proves that the corresponding electrolytic process does not depend on the remaining peaks.

All the processes described can be summarized by the following Figure 2:

There are different possibilities regarding the nature of L and L′, but the most probable is to associate the first form with the existence of conjugated double bonds systems as show in Figure 3, because this feature would favour the adsorption of these species on the electrode surface, which would justify the appearance of peak I_a_. Figure 3 represents the most probable scheme of the conversion reaction of L′ in to L.

Similar to systems I and II, the peaks are almost certainly related to the same type of electrochemical reactions in which the initial reactive species are slightly different. Such a difference would be the same as that gives rise to the existence of systems I and II, as evidenced by the fact that in which this splitting occurs, it also takes place from peak III in III and IV. It is expected that the electrochemical reaction associated with these peaks leads to the formation of species such as SCN^−^, S_8_, SO_4_^2−^, or some type of cyclization, detected in many cases by X-ray diffraction [34,35].

The more important parameters to estimate the reduction potentials of Cu^2+^ are the potentials of the peak (E_p_) or semipeak (E_p/2_) of I_c_ or II_c_ (Table 3). For the reversible process, the relationship between potential type follows the expression below:$$|E_{p}-E_{p/2}|=2.2\cdot \frac{R\cdot T}{n\cdot F}$$

If *n* = 1 and the temperature is 20°, |E_p_ − E_p/2_| has a value of 0.056 V.

In these conditions, the formal potential value of the Cu^2+^/Cu^1+^ (E°′) system can be calculated through the expression below:E°′=Ep+R⋅Tn⋅F·ln DoxDRe+1.109·R⋅Tn⋅F

### 2.8. DFT Study

This theoretical study undertakes an energetic analysis of hydrogen-bonded synthons as illustrated in Figure 2 (Compound **3a**), Figure 3 (Compound **4**), and Figure 5 (Compound **6**). Initially, the molecular electrostatic potential (MEP) surface of Compound **6** was computed. We have only analyzed the Cu-complex due to its neutrality, which facilitates a more accurate comparison of the hydrogen-bond donor/acceptor capabilities of the CO and NH groups of the barbituric ring and thiosemicarbazide group. The other two compounds, being salts, exhibit MEP values that are positive on the entire surface, due to the overall positive charge of the [Cu(HAcbDM)(H_2_O)_2_]^+^ fragment.

The MEP surface, showcased in Figure 7, identifies the MEP minimum at the chlorido ligand (−45.8 kcal/mol), indicating a negative MEP surface surrounding the Cl and extending to the Cu-coordinated O-atom. The carboxylic O-atoms of the barbituric ring exhibit MEP values of −38.3 kcal/mol and −30.4 kcal/mol, (O13 and O15, respectively, see Figure 1 for numbering scheme) highlighting their strong hydrogen-bond acceptor potential. In contrast, the MEP value at the S-atom is considerably lower (−12.5 kcal/mol). The highest MEP is observed at the NH group of the thiosemicarbazide group (64.4 kcal/mol), establishing it as the stronger hydrogen-bond donor. Additionally, the NH groups of the barbituric ring demonstrate significantly positive MEP values (30.9 and 33.6 kcal/mol for N15–H and N11–H, respectively), supporting the expectation of strong hydrogen-bond formation. This MEP analysis corroborates the formation of the diverse hydrogen-bonded synthons detailed previously.

To evaluate the hydrogen-bond (H-bond) energies in the assemblies depicted in Figure 8, Figure 9 and Figure 10, we employed the Quantum Theory of Atoms in Molecules (QTAIM) approach, calculating the strength of each H-bond using the method proposed by Espinosa et al. [36], which is based on electron density (ρ) at the bond critical point (CP) and uses the formula E = 0.5 × V. This method is particularly apt for the systems under consideration (**3a** and **4** are salts), where electrostatic interactions are predominant, allowing us to estimate H-bond contributions distinctly from ion-pair electrostatic forces.

For Compound **3a**, the tetrameric assembly in Figure 8 features an $R_{3}^{2}$(11) and two distinct $R_{2}^{2}$(8). Topological analysis highlighted each H-bond with a bond critical point (BCP, illustrated by fuchsia sphere) and a dashed bond indicating the bond path from the H-atom to the O-atom. The strengths of these interactions are denoted in blue, revealing the $R_{3}^{2}$(11) as the strongest (–16.1 kcal/mol), consistent with its formation of three H-bonds. The $R_{2}^{2}$(8) synthons also showed significant interaction energies (–15.4 and –12.1 kcal/mol), noteworthy for forming only two H-bonds. The energy variation between the synthons correlates with experimental H-bond lengths, being longer for the $R_{2}^{2}$(8) synthon linking two symmetrically equivalent [Cu(HAcbDM)(H_2_O)_2_]^+^ fragments (–12.1 kcal/mol).

Similarly, a trimeric assembly of Compound **4** (see Figure 9) has been studied that includes the three synthons crucial for forming ladder-like cation chains along the *c*-axis (see also Figure 3). The $R_{2}^{2}$(8) synthon associated with the barbituric rings has a formation energy of –8.2 kcal/mol, lower than those in Compound **3a**, probably because the carbonyl group also participates in CO···H_2_O bonds with the Cu-coordinated water molecule of an adjacent unit within the fused $R_{3}^{2}$(10) synthon. This latter synthon is the strongest one (–18.3 kcal/mol) due to the formation of three H-bonds. The $R_{2}^{2}$(8) synthon involving Cu-coordinated water molecules exhibits a significant formation energy, attributed to the enhanced acidity of the water protons when coordinated to the Cu atom.

In the case of Compound **6**, we focused on a dimeric assembly extracted from the zig-zag molecular chains along the “b” axis, as depicted in Figure 5. These chains extend through the formation of $R_{2}^{2}$(10) synthons. Our QTAIM analysis on this dimer, shown in Figure 10, highlighted that the $R_{2}^{2}$(10) synthon essentially comprises two interconnected $R_{2}^{2}$(8) and $R_{1}^{2}$(4) synthons, due to the NH bond forming a bifurcated NH···O,Cl hydrogen bond. The computed formation energies are –5.0 kcal/mol for the $R_{2}^{2}$(8) synthon and –2.3 kcal/mol for the $R_{1}^{2}$(4) synthon, indicating these as the less energetically favorable interactions compared to the other compounds studied in this manuscript.

## 3. Materials and Methods

### 3.1. Physical Measurements

All reagents and solvents were purchased from Sigma-Aldrich (Sigma-Aldrich. Inc., Tres Cantos, Madrid, Spain) and used as received, without further purification. Elemental analyses were performed with a Fisons-Carlo Erba 1108 microanalyser (CARLO ERBA Reagents SAS, Chaussée du Vexin, Val-de-Reuil, France). ^1^H and ^13^C NMR spectra were obtained as DMSO-d_6_ solutions with a Varian Mercury 300 instrument (Varian Medical Systems, Inc, Palo Alto, CA, USA). IR spectra were recorded as KBr disks (4000–400 cm^−1^) or polyethylene-sandwiched Nujol mulls (500–100 cm^−1^) with a Bruker IFS-66v spectrophotometer (Bruker Corporation, Billerica, MA, USA). Mass spectra were obtained in a HP5988A spectrometer for EI and a Micromass AUTOSPEC spectrometer (nitrobenzyl alcohol matrix) for FAB (Agilent Technologies, Inc., Santa Clara, CA, USA). Electronic spectra were carried out on a SHIMADZU UV-3101PC spectrophotometer (Izasa Scientific, Barcelona, Spain) equipped with a reflectance accessory. X-band EPR spectra of the complexes were obtained in 3 mm Pyrex tubes with a Bruker EMX spectrometer (Bruker Corporation, Billerica, MA, USA) using a conventional Dewar insert at liquid nitrogen temperature. Room temperature magnetic moments were determined on a PPMS (Physical Propierty Mesurement System) magnetometer Quantum Design, model SQUID (Quantum Design, San Diego, CA, USA). The conductivity measurements were carried out with a WTW conductivymeter model LF3 using freshly prepared 10^−3^ solutions of the complexes in DMF. Cyclic voltammograms were obtained on a 273 EG&G Princeton Applied Research electrochemical analyser (Artisan Technology Group, Champaign, IL, USA). In the measurements, DMF was used as the solvent and tetra(*n*-butyl)ammonium perchlorate (TBAP) as the supporting electrolyte.

### 3.2. Synthesis of Thiosemicarbazone Ligands

The ligands, 5-acetylbarbituric-4N-dimethylthiosemicarbazone, H_2_AcbDM, and 5-acetylbarbituric-4N-hexamethyleneiminyl thiosemicarbazone, H_2_AcbHexim·, were synthesized in excellent yields following our reported method [14,16], by condensation reactions between 5-acetylbarbituric acid [37] and 4-dimethyl-thiosemicarbazide or hexamethyleneiminyl thiosemicarbazide.

### 3.3. Synthesis and Crystallization of Complexes

An ethanolic suspension (20 mL) of thiosemicarbazone was added to a stirred solution of corresponding copper(II) salt in ethanol (15 mL) in molar ratio 1:1. The mixture was stirred at room temperature for a further 7 days. The resulting suspension was filtered off and the solid was washed with ethanol and dried over calcium chloride.

*[Cu(HAcbDM)(OAc)]·3H_2_O* (**1**·3H_2_O): Yield: 0.22 g (70.8%), m.p. 280 °C. Elemental analysis: Found: C, 29.5; H, 4.1; N, 15.8; S, 8.0. Calc. for C_11_H_21_O_8_N_5_SCu: C, 29.5; H, 4.7; N, 15.7; S, 7.2%. IR (ν_max_/cm^−1^): 3178, 3053 ν(NH); 1717, 1637 ν(C=O) + δ(NH); 1591, 1529 ν(CN + CC); 1295, 1268 ν(CS + CC); 1040 ν(NN); 810 ν(CS); 425 ν(Cu-N); 572, 527 ν(Cu-O); 312 ν(Cu-S). FAB^+^ MS [*m*/*z*, assignment]: 334 (3), [ML]^+^; 272 (2) [L]^+^. UV–Vis (*λ*_max_, cm^−1^): 28,409, 25,445, 17,544, 14,184. EPR (X-band, solid sample): *g*_av_ = 2.10. *μ*_eff_ = 2.07 MB. Conductance (Ohm^−1^·cm^2^**·**mol^−1^) in DMF: 12.*[Cu(HAcbDM)Cl]·1/2EtOH* (**2**·1/2EtOH): ·Yield: 0.20 g (76.7%), m.p. 240 °C. Elemental analysis: Found: C, 30.8; H, 3.5; N, 18.2; S, 7.6. Calc. for C_10_H_15_ClO_3.5_N_5_SCu: C, 30.6; H, 3.8; N, 17.8; S, 8.2%. IR (ν_max_/cm^−1^): 3156, 3084 ν(NH); 1711, 1642 ν(C=O) + δ(NH); 1583 ν(CN+ CC); 1306, 1241 ν(CS+ CC); 1060 ν(NN); 808 ν(CS); 449 ν(Cu-N); 553 ν(Cu-O); 314 ν(Cu-S). FAB^+^ MS [*m*/*z*, assignment]: 334 (9), [ML]^+^; 272 (2) [L]^+^. UV–Vis (*λ*_max_, cm^−1^): 29,326, 16,502, 14,420. EPR (X-band, solid sample): g_II_ = 2.27, *g*_⊥_ = 2.06, *g_av_* = 2.12, G = 4.12. *μ*_eff_ = 1.85 MB. Conductance (Ohm^−1^**·**cm^2^**·**mol^−1^) in DMF: 24.*[Cu(HAcbDM)(NO_3_)]* (**3**): Yield: 0.16 g (58.9%), m.p. 260 °C. Elemental analysis: Found: C, 27.8; H, 3.4; N, 20.6; S, 8.3. Calc. for C_9_H_12_O_6_N_6_SCu: C, 27.3; H, 3.1; N, 21.2; S, 8.1%. IR (ν_max_/cm^−1^): 3402–3190 ν(NH); 1709, 1643 ν(C=O) + δ(NH); 1586, 1512, ν(CN+ CC); 1297, 1242 ν(CS+ CC); 1038 ν(NN); 807 ν(CS); 1383, 1297 ν(NO_3_^−^); 425 ν(Cu-N); 552 ν(Cu-O); 312 ν(Cu-S). FAB^+^ MS [*m*/*z*, assignment]: 334 (9), [ML]^+^; 272 (2) [L]^+^. UV–Vis (*λ*_max_, cm^−1^): 29,325, 21,786, 17,544, 15,649, 14,184. EPR (X-band, solid sample): *g*_II_ = 2.19, *g*_⊥_ = 2.04, *g*_av_ = 2.06, G = 4.39. *μ*_eff_ = 2.14 MB. Conductance (Ohm^−1^**·**cm^2^**·**mol^−1^) in DMF: 54.

Suitable green crystals for X-ray diffraction were grown at room temperature as [Cu(HAcbDM)(H_2_O)_2_](NO_3_)·H_2_O (**3a**), after two weeks, from the resulting solution by slow evaporation at room temperature.

*[Cu(HAcbDM)(H_2_O)_2_]ClO_4_* (**4**): Yield: 0.15 g (45.6%), m.p. >300 °C. Elemental analysis: Found: C, 23.4; H, 3.5; N, 14.9; S, 6.1. Calc. for C_9_H_16_ClO_9_N_5_SCu: C, 23.1; H, 3.4; N, 14.9; S, 6.8%. IR (ν_max_/cm^−1^): 3398–3159 ν(NH); 1701–1640 ν(C=O) + δ(NH); 1579 ν(CN+ CC); 1308–1240 ν(CS+ CC); 811 ν(CS); 1121–1088, 627 ν(ClO_4_^−^); 446 ν(Cu-N); 553 ν(Cu-O). FAB^+^ MS [*m*/*z*, assignment]: 334 (42), [ML]^+^. EPR (X-band, solid sample): *g*_II_ = 2.20, *g*_⊥_ = 2.05, *g_av_* = 2.06, G = 4.34. *μ*_eff_ = 2.25 MB. Conductance (Ohm^−1^**·**cm^2^**·**mol^−1^) in DMF: 83.

Single crystals suitable for X-ray diffraction were obtained as (**4**), from the resulting solution by slow evaporation at room temperature.

*[Cu(HAcbHexim)(OAc)]·5H_2_O* (**5**·5H_2_O): Yield: 0.26 g (82.6%), m.p. 280 °C. Elemental analysis: Found: C, 33.6; H, 5.5; N, 13.5; S, 5.1. Calc. for C_15_H_31_O_10_N_5_SCu: C, 33.6; H, 5.8; N, 13.0; S, 5.9%. IR (ν_max_/cm^−1^): 3429–3042 ν(NH); 1702, 1671 ν(C=O) + δ(NH); 1587 ν(CN+ CC); 1297, 1245 ν(CS+ CC); 1031 ν(NN); 812 ν(CS); 433 ν(Cu-N); 559, 528 ν(Cu-O); 321 ν(Cu-S). FAB^+^ MS [*m*/*z*, assignment]: 437 (2) [ML]^+^; 326 (2) [L]^+^. UV–Vis (*λ*_max_, cm^−1^): 23,337, 17,637. EPR (X-band, solid sample): *g*_av_ = 2.08. *μ*_eff_ = 2.11 MB. Conductance (Ohm^−1^**·**cm^2^**·**mol^−1^) in DMF: 11.*[Cu(HAcbHexim)Cl]·1/2EtOH* (**6**·1/2EtOH): Yield: 0.12 g (47.0%), m.p. >300 °C. Elemental analysis: Found: C, 38.1; H, 4.3; N, 16.2; S, 6.9. Calc. for C_14_H_20.5_ClO_3.5_N_5_SCu: C, 37.7; H, 4.7; N, 15.7; S, 7.2%. IR (ν_max_/cm^−1^): 3257 ν(NH); 1710, 1653 ν(C=O) + δ(NH); 1589, 1560 ν(CN+ CC); 1309–1241 ν(CS+ CC); 1032 ν(NN); 808 ν(CS); 433 ν(Cu-N); 556 ν(Cu-O); 317 ν(Cu-S); 244 ν(Cu-Cl). FAB^+^ MS [*m*/*z*, assignment]: 388 (32), [ML]^+^; 326 (2) [L]^+^. UV–Vis (*λ*_max_, cm^−1^): 30,395, 21,954, 16,260, 15,408, 14,368. EPR (X-band, solid sample): *g*_av_ = 2.08. *μ*_eff_ = 1.89 MB. Conductance (Ohm^−1^**·**cm^2^**·**mol^−1^) in DMF: 35.

Single brown crystals suitable for X-ray diffraction were obtained as (**6**), from the resulting solution by slow evaporation at room temperature.

*[Cu(HAcbHexim)(NO_3_)]·1/2H_2_O* (**7**·1/2H_2_O): Yield: 0.04 g (16.7%), m.p. >300 °C. Elemental analysis: Found: C, 34.2; H, 4.2; N, 18.0; S, 6.3. Calc. for C_13_H_19_O_6.5_N_6_SCu: C, 34.0; H, 4.2; N, 18.3; S, 6.9%. IR (ν_max_/cm^−1^): 3345, 3185 ν(NH); 1711–1619 ν(C=O) + δ(NH); 1590, 1565 ν(CN+ CC); 1311–1247 ν(CS+ CC); 1092 ν(NN); 799 ν(CS); 1384, 1247 ν(NO_3_^−^); 433 ν(Cu-N); 532 ν(Cu-O); 322 ν(Cu-S). FAB^+^ MS [*m*/*z*, assignment]: 388 (4), [ML]^+^; 336 (2) [L]^+^. UV-vis (*λ*_max_, cm^−1^): 29,586, 22,075, 17,391, 15,385. EPR (X-band, solid sample): *g*_av_ = 2.11. *μ*_eff_ = 2.01 MB. Conductance (Ohm^−1^**·**cm^2^**·**mol^−1^) in DMF: 20.*[Cu(HAcbHexim)ClO_4_]* (**8**): Yield: 0.06 g (21.1%), m.p. >300 °C. Elemental analysis: Found: C, 32.3; H, 4.1; N, 14.0; S, 6.2. Calc. for C_13_H_18_ClO_7_N_5_SCu: C, 32.0; H, 3.7; N, 14.4; S, 6.6%. IR (ν_max_/cm^−1^): 3467–3037 ν(NH); 1711, 1652 ν(C=O) + δ(NH); 1590, 1561 ν(CN+ CC); 1309–1245 ν(CS+ CC); 1030 ν(NN); 809 ν(CS); 1121–1094, 677 ν(ClO_4_^−^); 441 ν(Cu-N); 555 ν(Cu-O); 316 ν(Cu-S). FAB^+^ MS [*m*/*z*, assignment]: 388 (8), [ML]^+^. UV–Vis (*λ*_max_, cm^−1^): 29,940, 22,727, 15,468, 14,409. EPR (X-band, solid sample): g_II_ = 2.37, *g*_⊥_ = 2.09, *g*_av_ = 2.17. *μ*_eff_ = 1.51 MB. Conductance (Ohm^−1^**·**cm^2^**·**mol^−1^) in DMF: 35.

### 3.4. Single-Crystal X-ray Diffraction

Diffraction data were obtained using an Enraf Nonius MACH3 automatic diffractometer from crystals mounted on glass fibers. Corrections for Lorentz and polarization effects, as well as absorption, were applied using a multi-scan method [38]. The structures were solved by direct methods [38], which revealed the positions of all non-hydrogen atoms. These were refined on *F*^2^ by a full-matrix least-squares procedure using anisotropic displacement parameters [39]. Hydrogen atoms were located in the difference maps and the positions of O–H and N–H hydrogen atoms were refined (others were included as riders). Isotropic displacement parameters of H atoms were constrained to 1.2/1.5 *U*_eq_ of the carrier atoms. Molecular graphics were generated with DIAMOND software (Version 4.6.2) [40]. The crystal data, experimental procedures, and refinement outcomes are summarized in Table 1.

### 3.5. Theoretical Methods

DFT calculations were performed using Gaussian-16 [41] software at the PBE0-D3/def2-TZVP level of theory [42,43,44]. The analysis focused on solid-state interactions utilizing crystallographic coordinates, where only the position of the H-atoms was optimized. Bader’s “Atoms in molecules” (QTAIM) theory [45] was employed to investigate hydrogen-bonding interactions through the AIMAll calculation package [46]. To estimate the H-bond energies we used the methodology proposed by Espinosa et al. [36]. The choice of the Espinosa equation for estimating the energy of hydrogen bonds in our study was driven by its simplicity and the robustness of its application in quantifying electrostatic interactions at BCPs, as detailed in [36]. While several other models exist for evaluating hydrogen bonds, this equation has been extensively validated against experimental and higher-level theoretical data, offering a reliable balance between computational efficiency and accuracy. Molecular electrostatic potential (MEP) surfaces (with an isosurface value of 0.001 atomic units) were generated with Gaussian-16. This methodology and level of theory has been previously used to investigate similar interactions [47,48]. Moreover, Rao et al. [49] evaluated the performance of various DFT methods in describing hydrogen-bond interactions, demonstrating that the PBE0 method provides the most accurate estimates of hydrogen-bond strengths.

## 4. Conclusions

In this study, we synthesized eight new copper(II) complexes (**1**–**8**) with the ligands 5-acetylbarbituric-4-diethylthiosemicarbazone (H_2_AcbDM) and 5-acetylbarbituric-4N-hexamethyleneiminyl thiosemicarbazone (H_2_Acbhexim). The complexes were characterized by spectroscopy, and the crystal structures of three of them, [Cu(HAcbDM)(H_2_O)_2_](NO_3_)·H_2_O (**3a**), [Cu(HAcbDM)(H_2_O)_2_]ClO_4_ (**4**), and [Cu(HAcbHexim)Cl] (**6**), were determined by single-crystal X-ray diffraction. All complexes are monomers and not electrolytes, except [Cu(HAcbDM)(H_2_O)_2_]ClO_4_, which is a 1:1 electrolyte, and the nitrate derivative with H_2_AcbDM, which is possibly a mixture of [Cu(HAcbDM)(NO_3_)] (neutral) and [Cu(HAcbDM)(H_2_O)_2_](NO_3_)·H_2_O (1:1 electrolyte). In the complexes, thiosemicarbazone ligands coordinate to copper(II) in a tridentate ONS manner and in a monoanionic tiona-keto mode.

The hydrogen bonds present in the crystal structures of Complexes **3a**, **4**, and **6** (N–H···O, O–H···O, C–H···O, and N–H···Cl in their case), as well as stacking interactions π–π in the second, led to the stabilization of the 3D crystal packing characterized by the presence of several supramolecular synthons, among which the graph set $R_{2}^{2}$(8) stands out. DFT calculations and MEP surface and QTAIM analyses emphasize the relevance of hydrogen-bonding interactions and the relative importance of the different H-bonded synthons. We shed light on the energetic characteristics of these interactions and disclosed that the $R_{2}^{2}$(8) synthon where only coordinated water molecules participate is the strongest one.

The EPR spectra of Complexes **1** and **5**–**7** show a single isotropic signal with an average g_iso_ value of 2.094, and Complexes **2**–**4** and **8** have two different g-values (average 2.258 (g_II_) and 2.063 (g_⊥_) indicating axial symmetry, and as g_II_ > g_⊥_ the presence of free electrons in the ground state d_x2−y2_ of a square-planar geometry is confirmed. The fact that most complexes of the H_2_AcbDM ligand have an isotropic signal and those of the H_2_Acbhexim ligand are anisotropic may be related to the volume of the substituent on the thioamide nitrogen atom.

Systematically, cyclic voltammograms of Complexes **1**–**4** and **6**–**8** were performed over a wide potential range in order to show all peaks. Measurements were then made in the potential range that bounded the zones where the individual peaks appeared to investigate possible connections between the associated electrochemical reactions. Accordingly, the redox activity of the complexes is attributed to the metal center. The cyclic voltammograms of all of them at different sweep velocities illustrate a linear relationship between the anodic peak currents and the square root of the sweeping velocities, confirming a quasi-reversible diffusion-controlled monoelectronic process of the Cu(II)/Cu(I) redox pair.

## Data Availability

The data presented in this study are available in article and Appendix A.

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
