# Peer review of "Synthesis, Structural Characterisation, and Electrochemical Properties of Copper(II) Complexes with Functionalized Thiosemicarbazones Derived from 5-Acetylbarbituric Acid"

_molecules, 2024, doi:10.3390/molecules29102245_

Round 1

Reviewer 1 Report

Comments and Suggestions for Authors

Dear editor,

     The manuscript reported the synthesis, structural characterization and electrochemical properties of a series of copper(II) complexes with Functionalized Thiosemi-carbazones derived from 5-Acetylbarbituric Acid. The manuscript is well written, also the data are sound, I recommend the publication of the manuscript. However, some problems of the manuscript should be revised before its publication.

1.       Only three of these reported compounds are structurally characterized, they are 3a, 4 and 6, but in the section of 2.1, there is compound 3, what is the difference between the compounds 3 and 3a, please make a careful discussion. Also in this section, the authors should emphasis that only three of these compounds are structurally characterized and why the other compounds were not structurally characterized?

2.       In the EPR spectra, 2-4 and 8 present the characteristics of axial anisotropy with well-defined values of g×€×€ and g, why the spectra of compounds 3-6 are present in the manuscript?

3.       The manuscript is too long, some descriptions or even some sections can be removed to the supplementary material.  

Reviewer 2 Report

Comments and Suggestions for Authors

The present study introduced new copper(II) complexes with the ligands 5-acetylbarbituric-4-diethylthiosemicarbazone and 5-acetylbarbituric-4N-hexamethyleneiminyl thiosemicarbazone. The structure of these complexes was fully characterized by such spectral and analytical methods as X-ray diffraction, UV, ESI-HRMS, IR, EPR and cyclic voltammetry. Also, using theoretical calculations, some energy characteristics of hydrogen bonds of the synthesized complexes were established.

Despite the fact that within the framework of this study, comprehensive and extensive work on structural research was carried out, the article lacks fundamentality. Only a number of 8 essentially homologous complexes have been studied, only their partial structural analysis is carried out, there is no general novelty and no global patterns have been established.

Nevertheless, the manuscript is written in quite literate language, and the course of the research is quite logical. There are several points that need to be corrected:

- line 453: (O13 and O13 respectively, see Figure 3... - perhaps there was a typo in the numbering of atoms?

- why was this particular computational scheme of PBE0-D3/def2-TZVP used? The answer must be justified, since the text deals with numbers up to ~2 kcal/mol. This means that using an another functional and/or basis may lead to result with an otherwise order of significant digits.

- why was the Espinosa equation used to estimate the energy of H-bonds? At the moment, several dozen other, no less (and perhaps even more) adequate equations for evaluating hydrogen bonds are known, perhaps more suitable for this particular case. This needs to be discussed in more detail. I understand that you have used the Espinosa equation previously in your studies when evaluating similar fragments, but this is not an argument, a discussion is necessary.

After the comments are eliminated, the article can be accepted for publication.

Comments on the Quality of English Language

English needs to be corrected, sometimes the text becomes too colloquial.

Reviewer 3 Report

Comments and Suggestions for Authors

The manuscript describes the synthesis of 8 copper complexes of functionalized thiosemicarbazones derived from 5-acetyl barbituric acid. The authors have carried out a variety of characterization techniques that improves the scientific soundness of the manuscript. The manuscript can be accepted for publication after major revisions. The specific suggestions are detailed below:

1) The lengthy introduction part is too difficult to follow. Hence it is recommended to cut short the introduction.

2) The structure of synthesized copper complexes [Cu(HAcbDM)(OAc)]·3H2O (1·3H2O), [Cu(HAcbDM)Cl]·1/2EtOH(2·1/2EtOH), [Cu(HAcbDM)(NO3)] (3), [Cu(HAcbDM)(H2O)2]ClO4 (4), [Cu(HAc-bHexim)(OAc)]·5H2O (5·5H2O), [Cu(HAcbHexim)Cl]·1/2EtOH (6·1/2EtOH), [Cu(HAc-bHexim)(NO3)]·1/2H2O (7·1/2H2O) and [Cu(HAcbHexim)ClO4] are not seen anywhere in the manuscript in its original form. It should be included at the results and discussion section.

3) I would like to know the applications of these complexes. Whether the authors have carried out any studies in this area?

After addressing these comments, the manuscript may be published.

Comments on the Quality of English Language

Minor editing of English language is recommended.
